# Microbiota Analysis and Characterisation of the Novel *Limosilactobacillus* Strains Isolated from Dogs

**DOI:** 10.3390/microorganisms13051059

**Published:** 2025-05-01

**Authors:** Ga-Yeong Lee, Hae-Yeon Jo, Muhammad Aleem Abbas, Ling Gui, Md Sekendar Ali, Seung-Jun Kim, Seung-Chun Park

**Affiliations:** 1Laboratory of Veterinary Pharmacokinetics and Pharmacodynamics, Institute for Veterinary Biomedical Science, College of Veterinary Medicine, Kyungpook National University, Daegu 41566, Republic of Korea; yeong1129@knu.ac.kr (G.-Y.L.); whgodus@knu.ac.kr (H.-Y.J.); syedaleemabbas77@gmail.com (M.A.A.); guiling@knu.ac.kr (L.G.); alipharm2000@gmail.com (M.S.A.); 2Veterinary Obstetrics, Institute for Veterinary Biomedical Science, College of Veterinary Medicine, Kyungpook National University, Daegu 41566, Republic of Korea; 3Cardiovascular Research Institute, Kyungpook National University, Daegu 41566, Republic of Korea

**Keywords:** dogs, lactic acid bacteria, microbiota, probiotic

## Abstract

**Background/Objectives:** The growing number of households with companion dogs raises concerns. Co-living environments between companion dogs and their owners are linked to a heightened risk of cross-infections from strains such as *Escherichia coli* (*E. coli*), *Staphylococcus aureus* (*S. aureus*), *Salmonella*, and faecal coliforms. Therefore, this study aims to propose measures for healthy cohabitation by analysing the faecal microbiota of puppies and adult dogs. **Methods:** We isolated lactic acid bacteria (LAB) from their faeces and assessed their potential to inhibit *E. coli*, *S. aureus*, and *Salmonella* spp. Faecal samples from puppies (<2 months old) and adult dogs (>12 months old) were analysed and compared. **Results:** The analysis revealed that *Lactobacillus* dominated puppy faeces, while *Bacteroidetes* were more abundant in adult dogs. In total, 109 primary LAB candidates were isolated from faecal samples. These isolates underwent secondary screening for acid tolerance, bile salt resistance, acid production, heat resistance, protease activity, and antimicrobial activity against *E. coli*, *S. aureus*, and *Salmonella* spp. Five secondary LAB candidates with probiotic potential were further characterised via morphological and genetic analysis. All five strains were *Lactobacillus reuteri*, with *L. reuteri* JJ37, JJ68, JJ69, JJ71, and JJ77 emerging as the final probiotic candidates. **Conclusions:** They promote healthier cohabitation between dogs and their owners.

## 1. Introduction

Gastrointestinal health in companion animals, particularly dogs, remains a critical concern for veterinarians and pet owners worldwide. *Salmonella* spp. and *Escherichia coli* (*E. coli*) are among the most significant bacterial pathogens affecting canine gut health, often causing severe gastroenteritis and systemic infections, posing risks of zoonotic transmission [1]. Conventionally, these infections have been managed with antibiotics, but the increasing prevalence of antimicrobial resistance has necessitated alternative therapeutic approaches. In response, the World Health Organization has advocated for restricted antibiotic use since 1997, recommending administration only in cases of definitive bacterial infections. Excessive antibiotic use not only eradicates pathogens but also disrupts beneficial gut microbiota, leading to dysbiosis and secondary health complications [2]. Furthermore, the emergence of extended-spectrum β-lactamase-producing *E. coli* in dogs and cats highlights the urgent need for alternative infection management approaches that minimise antibiotic resistance risks [3,4].

Among these alternatives, probiotics have emerged as a promising strategy for maintaining gut health and preventing infections in dogs. *Lactobacillus* species, in particular, enhance gut microbiota composition, support immune function, and inhibit pathogenic bacterial growth [5]. A study shows that probiotic supplementation improves microbial diversity, increases beneficial bacteria, and suppresses pathogenic pathways, particularly in dogs with gastrointestinal disturbances such as diarrhoea [6]. Additionally, probiotics mitigate antibiotic-induced dysbiosis, thereby preserving gut microbial homeostasis [7]. Therefore, maintaining the gut microbiome using probiotics could provide significant benefits in managing gastrointestinal diseases and promoting overall canine health.

Beyond gastrointestinal health, probiotics offer potential benefits in systemic health conditions, such as chronic kidney disease, where they help maintain nutritional status and improve renal function parameters [8]. The synergistic effects of probiotics, prebiotics, and antioxidants have been explored as therapeutic interventions for chronic diseases in dogs, further underscoring the broad applications of probiotics in veterinary medicine. Moreover, as conventional antibiotic therapy remains limited, the growing demand for alternative treatments, including targeted probiotic formulations, emphasises the need for continued research into their specific mechanisms and benefits, representing a critical advancement in the animal medicine sector [6]. The ability of *Lactobacillus* strains to inhibit the growth of *Salmonella* and *E. coli* highlights their potential as effective probiotics for canine gastrointestinal health. Therefore, this study aims to identify and characterise *Lactobacillus* strains with antimicrobial activity against key canine pathogens. A two-step screening process was employed: microbiome analysis in healthy puppies and adult dogs was conducted to isolate primary *Lactobacillus* candidate strains; secondly, these strains were evaluated for their probiotic potential, focusing on their antimicrobial activity against *Salmonella* and *E. coli*.

## 2. Materials and Methods

### 2.1. Chemicals and Reagents

To investigate the distribution of cultivable gut microbiota in canine faecal samples, various media were used. These included plate count agar (PCA) with Bromocresol Purple (BCP), transgalactosylated oligosaccharides (TOS)-MUP (TOS), potato dextrose agar (PDA), tryptone sulfite neomycin (TSN), MacConkey (MC), Salmonella–Shigella (SS), mannitol salt (MS), Brilliant Green (BG), and MacConkey sorbitol (MACS), all purchased from Difco BD (Franklin Lakes, NJ, USA). De Man, Rogosa, and Sharpe (MRS) and blood agar were obtained from Kisan Bio (Seoul, Republic of Korea), while Bacto Agar was purchased from Difco BD (Franklin Lakes, NJ, USA). For the acid resistance assay, 10 M HCl was obtained from DC Chemical Co., Ltd. (Seoul, Republic of Korea), and for the bile salt resistance assay, bile salts were purchased from Difco BD (Franklin Lakes, NJ, USA). Agarose for PCR was obtained from Yeong Science (Bucheon, Gyeonggi-do, Republic of Korea). All reagents used were of analytical grade or higher.

### 2.2. Faecal Sample Collection

To select candidate probiotic strains and analyse microbiomes, fresh faecal samples from healthy puppies and adult dogs were obtained from a Gyeongsan animal shelter (Gyeongsan, Gyeongsangbuk-do, Republic of Korea). The samples came from 30 healthy adult dogs (5–15 kg, mixed breed, 1–8 years old) and 10 puppies (1–2 kg ± 0.5 kg, mixed breed, 2–3 months old) administered full veterinary care (Appendix A). The fresh faecal samples were collected using a standardised protocol through induced defecation or rectal swab sampling in healthy dogs [9]. Figure 1 shows the process for selecting probiotic candidates from dogs.

### 2.3. DNA Extraction and 16S rRNA Gene Sequencing

Genomic DNA was extracted from the faecal samples of puppies and adult dogs using the G-spin^TM^ Genomic DNA Extraction Kit (Qiagen Inc., Hilden, Germany) following the instructions of the manufacturer. DNA quality control was performed on the filtered supernatant using a Take 3 plate with an Agilent microplate reader (Agilent, Santa Clara, CA, USA). A DNA concentration of at least 15 ng/µL and an A260/280 ratio of at least 1.8 were obtained to ensure sample quality. DNA extracts that passed quality control underwent PCR analysis to confirm their genetic profiles. For this purpose, the AccuPower^®^ ETEC-Toxin 4-Plex PCR kit (Bioneer, Daejeon, Republic of Korea) was used according to the instructions of the manufacturer. The purified DNA was quantified, and 1 ng was used as the template for PCR amplification. The V4 region of the 16S rRNA gene was amplified with universal primers (515F and 806R) by Macrogen (Macrogen Inc., Seoul, Republic of Korea). Next-generation sequencing was performed using the EzBioCloud MTP pipeline and the EzBioCloud 16S database PKSSU4.0 to analyse the microbiome taxonomic profile. For diversity and statistical analysis, normalised operational taxonomic units were utilised. The biodiversity (alpha-diversity) within samples was determined using Chao1 and Shannon indices, while the similarities between the samples (beta-diversity) were determined through weighted UniFrac calculations. Principal coordinate analysis was conducted via Curtis similarity clustering analysis.

### 2.4. Distribution of Microorganisms in Puppies and Adult Dogs and Primary Selection of Candidate Probiotic Strains

Fresh faecal samples from puppies and adult dogs were collected through induced defecation or rectal swabs. The samples were immediately placed in an icebox and transported to the laboratory under anaerobic conditions at temperatures below 4 °C. To isolate LAB, selective media such as PCA with BCP, MRS, and TOS-MUP (selective for *Bifidobacterium*) were used. Moreover, these selective media were also applied to analyse the gut microbiota of puppies and adult dogs.

We used blood agar to assess total microbial counts, while PDA was utilised to identify fungi and yeast. TSN was used for detecting anaerobic sulfite-reducing bacteria, MC for *E. coli*, and SS for isolating *Salmonella* and *Shigella* species. MS was used for the selective cultivation of *Staphylococcus* species, BG for *Salmonella* spp., and MACS for enterohaemorrhagic *E. coli*.

Samples were serially diluted 10-fold (10^−1^, 10^−3^, 10^−5^, and 10^−7^) using sterilised 0.1% agar saline and spread-plated onto selective media. Blood agar, PDA, TSN, MAC, MS, BG, MACS, and SS were incubated aerobically at 37 °C for 24 h. MRS, PCA with BCP, and TOS-MUP plates were incubated anaerobically in an Oxoid anaerobic jar with O_2_-free CO_2_ at 37 °C for 48 h. Colonies that developed on the media were selected as primary probiotic candidates following the Mitsuoka method [10] and stored at −20 °C for further analysis.

### 2.5. Secondary Selection of the Primary Candidate Strains

#### 2.5.1. Acid Tolerance Test

To identify candidate probiotic strains, the primary selection focused on acid-tolerant LAB. Candidate strains were initially cultured in MRS broth at 37 °C for 24 h. The cultured strains were then inoculated at 1% (*v*/*v*) into MRS broth and adjusted to pH 2 and pH 4 using 1 M HCl. Then, the bacteria were incubated with shaking for 1.5 h, and LAB survival was assessed by plating the cultures onto MRS agar plates and counting the resulting colonies. Strains that formed colonies under these conditions were considered acid-tolerant and selected as primary candidates.

#### 2.5.2. Lactic Acid Production Test

To identify acid-producing probiotic candidates, 119 LAB were isolated from 57 plates containing primary acid-tolerant strains. These isolates were inoculated into MRS broth and incubated at 37 °C for 24 h for initial cultivation. The cultured strains were then streaked onto MRS agar supplemented with 1% CaCO_3_ and incubated at 37 °C for 48–72 h. Strains that formed clear halos around their colonies were classified as acid-producing and selected as secondary candidates.

#### 2.5.3. Bile Tolerance Test

In total, 40 isolates identified as LAB through secondary screening were further assessed for bile tolerance. These isolates were inoculated in MRS broth and incubated at 37 °C for 24 h for primary cultivation. Subsequently, the candidate strains were streaked evenly onto the surface of MRS agar plates containing 1% oxgall (Difco^TM^ Oxgall, BD, Franklin Lakes, NJ, USA) using a sterilised loop and incubated at 37 °C for 48 h. Strains that formed colonies on the MRS agar plates were selected as bile-tolerant strains.

#### 2.5.4. Heat Tolerance Test

In the tertiary screening, to identify candidate probiotic strains with heat tolerance, 39 bile-tolerant isolates were tested. The candidate strains were inoculated into MRS broth and incubated at 37 °C for 24 h for primary cultivation. The primary cultures were then transferred into fresh MRS broth at a 1% inoculum ratio and subjected to shaking incubation at 40 °C, 50 °C, and 60 °C for 1 h. The survival of the strains was evaluated by streaking the cultures onto MRS agar plates and counting colony formations. Strains that formed colonies were selected as heat-tolerant for the quaternary screening.

#### 2.5.5. Selection of Protease-Producing Strains

In the quaternary screening, to identify protease-producing candidate probiotic strains, 28 heat-tolerant strains were analysed. The isolates were spotted onto MRS agar plates supplemented with 2% skim milk using a sterilised loop. The plates were then incubated at 37 °C for 24 h, and strains that formed a clear zone (halo) around their colonies were identified as protease-producing strains. *Lactobacillus acidophilus* (*L. acidophilus*) KCTC 3111 was used as the control strain.

#### 2.5.6. Selection of *Lactobacillus* spp. and Antibacterial Activity Against *E. coli* and *Salmonella*

The colony morphology, Gram staining, microscopic cell shape, and catalase activity (H_2_O_2_ test) were analysed. Strains demonstrating antibacterial activity against *E. coli* ATCC 35218, *Salmonella* Typhimurium KCTC 2515, and *Staphylococcus aureus* (*S. aureus*) ATCC 29,213 were selected as primary antibacterial candidates. A single colony from each strain was isolated from the plate and diluted in saline. Pathogenic strains were adjusted to 10^3^ CFU/mL, while candidate probiotic strains were adjusted to 10^8^ CFU/mL. *S. aureus* ATCC 29213 and *E. coli* ATCC 35218 were co-cultured with the candidate strains in 5 mL Iso-Sentitest–MRS broth (9:1), whereas *S.* Typhimurium KCTC 2515 was co-cultured in 5 mL 2 × MRS-MHB broth for 24 h at 37 °C. After incubation, 100 µL of each culture was sampled, serially diluted (10-fold), and plated on MHB agar (*S. aureus* ATCC 29213 and *E. coli* ATCC 35218) and SS agar (*S.* Typhimurium KCTC 2515). A 20 µL aliquot from each dilution was spread onto the respective agar plate and incubated at 37 °C for 24 h. For controls, cultures in Iso-Sentitest–MRS (9:1) or 2 × MRS-MHB broth without treatment served as negative controls, and cultures treated with gentamicin (0.1 mg/mL) were used as positive controls. Subsequently, the bacterial colonies were enumerated.

### 2.6. Identification of the Secondary Candidate Strains

#### Identification of *Lactobacillus* spp. JJ37, 68, 69, 71, and 77

The *Lactobacillus* spp. strains JJ37, 68, 69, 71, and 77 were identified through the screening process and confirmed through 16S rRNA gene sequencing. In brief, genomic DNA was extracted from the isolates using a Qiagen genomic DNA extraction kit (Qiagen Inc., Hilden, Germany). Consequently, polymerase chain reaction (PCR) was performed using the forward primer (5′-AGGTAACGGCTTACCAAGGC-3′) and the reverse primer (5′-CCACCGCTACACATGGAGTT-3′). The PCR mixture contained 50 pmol of primers, 50 ng of template DNA, 5 μL of 10 × Taq DNA polymerase buffer, and 1 U of Taq DNA polymerase (Takara, Japan). The PCR protocol included an initial denaturation at 94 °C for 5 min, followed by 35 cycles of denaturation at 95 °C for 30 s, annealing at 56 °C for 30 s, and elongation at 72 °C for 30 s, with a final extension step at 72 °C for 7 min. The resulting sequences were compared to closely related sequences in the GenBank database using BLAST software (version 2.8.1). A phylogenetic tree was created via the neighbor-joining method using MEGA (version 5.0) software.

### 2.7. Statistical Analysis

Statistical analyses, including data processing and graph creation, were performed using GraphPad Prism 8.0 (San Diego, CA, USA). One-way analysis of variance (ANOVA) followed by Tukey’s post hoc test was used for group comparisons. Results were expressed as mean ± standard error of the mean (SEM), with statistical significance set as *p* < 0.05.

## 3. Results

### 3.1. DNA Quality and Purity Assessment of Gut Microbiota for Metagenomic Analysis

The quality of DNA was verified through spectrophotometry, gel electrophoresis, and PCR amplification. Quality control parameters, including sample concentrations, volumes, and DNA purity, were assessed based on the following criteria: a minimum concentration of 15 ng/μL, volume of at least 20 μL, and DNA purity ratio (A260/280) within the range of 1.8–2.1. DNA purity was determined by calculating the absorbance ratio at 260 nm (indicative of DNA) to 280 nm (indicative of proteins). All samples (A2, A4, A10, B3, B5, and B7) met the quality control criteria and they were confirmed via PCR analysis (Appendix A). The successful amplification of the target DNA in each sample was validated by observing the presence of distinct bands on the gel (Appendix A).

### 3.2. Metagenomic Analysis of the Gut Microbiota in Puppies and Adult Dogs

Figure 2 illustrates the comparison of the gut microbiota composition between healthy puppies and adult dogs. At the phylum level, puppy B3 showed the highest proportion of *Proteobacteria* (78.18%), while puppy B5 was dominated by *Firmicutes* (62.03%) and puppy B7 by *Fusobacteria* (62.08%). No single dominant phylum was shared among the puppies. In contrast, adult dogs A2, A4, and A10 showed a consistent prevalence of *Bacteroidetes* (>40%) across all samples. Overall, *Proteobacteria*, *Firmicutes*, *Fusobacteria,* and *Bacteriodetes* were the dominant phyla (>5%) in both puppies and adult dogs. However, no commonly dominant phylum was observed between the two groups.

At the class level, puppy B3 was dominated by *Gammaproteobacteria* (76.91%), followed by *Clostridia* (17.22%). Puppy B5 showed the highest proportion of *Negativicutes* (38.77%) and *Bacteroidia* (20.62%), while puppy B7’s sample was primarily composed of *Fusobacteria* (62.09%), followed by *Gammaproteobacteria* (15.81%). The predominant classes (>30%) in each puppy were *Gammaproteobacteria*, *Negativicutes*, and *Fusobacteria*, respectively, while no shared dominant class was observed among the puppies. In adult dogs, A2 was dominated by *Bacterodia* (59.48%) and *Clostridia* (20.34%), A4 by *Bacteroidia* (45.61%) and *Fusobacteria* (28.88%), and A10 by *Bacteroidia* (40.9%) and *Clostridia* (25.97%). A consistent dominance of *Bacteroidia* (>40%) was observed across all adult dogs, but no common dominant class was identified between puppies and adult dogs.

At the order level, puppy B3 had the highest proportion of *Enterobacterales* (75.33%), followed by *Clostridiales* (17.22%) and *Bacterodiales* (2.79%). Puppy B5’s sample was primarily composed of *Veillonellales* (36.96%), followed by *Bacteroidales* (20.62%) and *Lactobacillales* (10.28%). Puppy B7 showed the highest abundance of *Fusobacteriales* (62.09%), followed by *Pseudomonadale* (7.54%) and *Campylobacterales* (6.66%). Each puppy was dominated by different orders—*Enterobacterales*, *Veillonellales,* and *Fusobacteriales*, respectively—with no shared dominant order observed. In adult dogs, A2 exhibited the highest proportion of *Bacteroidales* (59.47%), followed by *Clostridiales* (20.34%) and *Fusobacteriales* (5.96%); A4 predominantly contained *Bacteroidales* (45.61%) followed by *Fusobacteriales* (28.86%) and *Aeromonadales* (6.2%); and A10 was dominated by *Bacterodiales* (40.9%) followed by *Clostridiales* (25.96%) and *Erysipelotichales* (8.57%). A consistent dominance of *Bacteroidales* (>40%) was observed across all adult dogs. However, similar to the phylum and class levels, no common dominant order was identified between puppies and adult dogs.

### 3.3. Differences in Gut Microbial Composition Between Puppies and Adult Dogs

The differences in gut microbiota composition between puppies and adult dogs were analysed by comparing correlations at the phylum, class, and order levels (Figure 3). At the phylum level, puppies showed higher relative abundances of *Firmicutes*, *Fusobacteria*, *Proteobacteria*, *Spirochaetes,* and *Verrucomicrobia*, while adult dogs had higher proportions of *Actinobacteria*, *Bacteroidetes*, *Cyanobacteria*, *Deferribacteres*, *Synergistetes*, *Tenericutes*, and *Thermotogae*. The phyla with more than a 1.5-fold difference between the groups included *Bacteroidetes*, *Fusobacteria,* and *Proteobacteria*. *Bacteroidetes* accounted for 7.98% of the microbiota in puppies and 48.66% in adult dogs, representing an approximately 6-fold difference. The *Fusobacteria* constituted 21.82% in puppies and 14.23% in adult dogs, a 1.53-fold difference. *Proteobacteria* comprised 6.49% in puppies and 38.21% in adult dogs, showing a 5.88-fold difference. No other phyla exhibited differences exceeding 1.5-fold.

At the class level, puppies showed higher relative abundances of *Bacilli*, *Bacteroidia*, *Betaproteobacteria*, *Clostridia*, *Epsilonproteobacteria*, *Erysipelotrichi*, *Fusobacteria_c*, *Gammaproteobacteria*, and *Negativicutes*. In contrast, adult dogs had higher proportions of *Bacteroidia*, *Betaproteobacteria*, *Clostridia*, *Erysipelotrichi*, *Fusobacteria_c*, *Gammaproteobacteria*, and *Negativicutes*. Classes with differences (>1.5 fold) between puppies and adult dogs included *Bacilli*, *Bacterodia*, *Clostridia*, *Epsilonproteobacteria*, *Fusobacteria_c*, *Gammaproteobacteria*, and *Negativicutes*. For instance, *Bacilli* represented 4.79% of the microbiota in puppies and 0.39% in adult dogs (12.28-fold difference). *Bacteroidia* accounted for 7.89% in puppies and 48.66% in adult dogs (6.09-fold difference). *Clostridia* was present at 9.46% in puppies and 17.45% in adult dogs (1.84-fold difference). *Epsilonproteobacteria* accounted for 3.14% in puppies and 5.25% in adult dogs (1.67-fold difference). *Fusobacteria_c* constituted 21.82% in puppies and 14.23% in adult dogs (1.53-fold difference). *Gammaproteobacteria* comprised 32.16% in puppies and 3.21% in adult dogs (10.01-fold difference). *Negativicutes* accounted for 13.01% in puppies and 5.24% in adult dogs (2.48-fold difference).

At the order level, puppies showed higher relative abundances of *Aeromonadales*, *Alteromonadales*, *Bacillales*, *Bacteroidales*, *Brachyspirales*, *Burkholderiales*, *Campylobacterales*, *Caulobacterales*, *Clostridiales*, *Enterobacterales*, *Erysipelotrichales*, *Fusobacteriales*, *Lactobacillales*, *Mycoplasmatales*, *Pseudomonadlaes*, *Spirochaetales*, *Sphingomonadales*, *Veillonellales*, *Verrucomicrobiales*, and *Vibrionales*. In contrast, adult dogs exhibited higher proportion of *Acidaminoccocales*, *Acidimicrobiales*, *Acholeplasmatales*, *Actinomycetales*, *Bacteroidales*, *Bifidobacteriales*, *Clostridiales*, *Chroococcales*, *Coriobacteriales*, *Corynebacteriales*, *Kosmotogales*, *Pasteurellales*, *Rhodocyclales*, *Selenomonadales*, *Synergistales*, and *Tissierellales*. Orders with differences (>1.5 fold) between puppies and adult dogs included *Acidaminococcales*, *Bacteroidales*, *Clostridiales*, *Enterobacterales*, *Fusobacteriales*, *Lactobacillales*, *Pseudomonadales*, and *Veillonellales*. For example, *Acidaminococcales* accounted for 0.15% of the microbiota in puppies and 4.05% in adult dogs (25.8-fold difference). *Clostridiales* was present at 9.46% in puppies and 17.45% in adult dogs (1.84-fold difference). *Enterobacterales* accounted for 26.06% in puppies and 14.23% in adult dogs (1.53-fold difference). *Lactobacillales* comprised 4.76% in puppies and 0.39% in adult dogs (12.24-fold difference). *Pseudomonadales* constituted 2.54% in puppies but were absent in adult dogs. *Veillonellales* accounted for 12.33% in puppies and 0.02% in adult dogs (456-fold difference). No other order exhibited a significant difference (>1.5 fold).

### 3.4. F/B Ratio in Adult Dogs and Puppies

To investigate the *Firmicutes*/*Bacteroidetes* (F/B) ratio and its potential effect, the gut microbiota composition in adult dogs and puppies was analysed. Genomic DNA was extracted, and 16S rRNA gene sequencing was performed to profile the gut microbiota. The relative abundances of *Firmicutes* and *Bacteroidetes* at the phylum level were compared. In puppies, *Firmicutes* accounted for 34.54% on average (range: 18.66–62.03%), while in adult dogs, it was 28.35% on average (range: 16.48–39.35%) Figure 4a. Conversely, *Bacteroidetes* was significantly less abundant in puppies, with an average of 7.99% (range: 0.55–20.63%), compared to that in adult dogs, which exhibited an average of 48.66% (range: 40.9–59.48%) (Figure 4b). These findings indicate that puppies had a higher proportion of *Firmicutes* and a lower proportion of *Bacteroidetes* than adult dogs (Figure 4). The F/B ratio was calculated as the relative abundance of *Firmicutes* divided by that of *Bacteroidetes*. The F/B ratio in puppies was 17.13 on average (range: 3.01–41.7), whereas in adult dogs, it was 0.6 (range: 0.36–0.96) (Figure 4c). This suggests that the F/B ratio is significantly higher in puppies than in adult dogs.

### 3.5. Differences in Gut Microbiota Diversity Between Puppies and Adult Dogs

In total, 263,479 V4 16S rRNA sequence reads were collected from six samples, with an average of 41,472 reads per sample. Species richness and diversity were assessed using ACE, Chao1, and Jackknife, Shannon, and Simpson indices. The ACE, Chao1, and Jackknife indices were significantly higher in adult dogs than in puppies (*p* < 0.05) (Figure 5).

### 3.6. Comparison of Faecal Microbial Distribution Between Puppies and Adult Dogs

To compare the faecal microbial distribution between healthy puppies and adult dogs, colony counts were obtained via direct culture using selective media. Figure 6 and Appendix A show the microbial distribution of faecal samples from 10 puppies and 30 adult dogs on each medium. On blood agar, the colony count for puppies was 8.63 log CFU/mL, while for adult dogs, it was 6.68 log CFU/mL. This showed that puppies had 29.2% more colonies than adult dogs (Figure 6a). On TSN, the colony count was 5.50 log CFU/mL for puppies and 5.04 log CFU/mL for adult dogs. This showed that puppies had approximately 9.1% more colonies than adult dogs (Figure 6b). On PDA, puppies had 6.94 log CFU/mL, and adult dogs had 6.36 log CFU/mL, showing that puppies had 9.1% more colonies than adult dogs (Figure 6c). On MAC, both puppies and adult dogs had colony counts of 4.5 log CFU/mL (Figure 6d). On SS, puppies had 1.76 log CFU/mL and adult dogs had 3.43 log CFU/mL, showing that adult dogs had 94.9% more colonies than puppies (Figure 6e). On MS, puppies had 7.39 log CFU/mL, and adult dogs had 3.3 log CFU/mL, confirming that puppies had 124% more colonies than adult dogs (Figure 6f). On BG, puppies had 6.64 log CFU/mL, and adult dogs had 5.36 log CFU/mL, showing that puppies had 23.9% more colonies than adult dogs (Figure 6g). On MACS, both puppies and adult dogs had colony counts of 4.5 log CFU/mL (Figure 6h). On MRS, colony counts were 7.45 log CFU/mL for puppies and 6.83 log CFU/mL for adult dogs (Figure 6i). On TOS, puppies and adult dogs had 6.42 log CFU/mL and 6.13 log CFU/mL, respectively (Figure 6j). On PCA, puppies had 8.29 log CFU/mL and adult dogs had 7.54 log CFU/mL (Figure 6k). Across PCA, MRS, and TOS media, puppies had approximately 10%, 9.1%, and 4.7% more colonies than adult dogs. No significant differences were observed between puppies and adult dogs in all media except for the blood agar and MS media.

### 3.7. Selection of Lactic Acid Bacteria Strains Through Multi-Step Screenings

#### 3.7.1. Acid Tolerance Test

Forty faecal samples from dogs were diluted 10-fold and spread on PCA with BCP agar to select 109 lactic-acid-producing strains. To identify strains with high probiotic potential, an acid tolerance test was performed. The survival rates at pH 2 and pH 4 were measured (Table 1 and Figure 7a). At pH 2, 59 out of 109 strains (54.12%) survived, while at pH 4, 107 out of 109 strains (98.17%) survived. The control strain, *L. acidophilus* KCTC 3111, survived at both pH levels, confirming its acid tolerance.

#### 3.7.2. Lactic Acid Production Test

A test was performed to isolate lactic-acid-producing strains. Among the 119 strains, 40 (33.61%) produced lactic acid, while 79 (66.39%) did not (Table 1 and Figure 7b). The lactic-acid-producing strains were categorised by colony size: small, medium, or large. Among the 40 strains, 30 produced the smallest colonies, 7 produced medium-sized colonies, and 3 produced the largest colonies. The control strain, *L. acidophilus* KCTC 3111, produced medium-sized colonies and was identified as a lactic-acid-producing strain.

#### 3.7.3. Bile Tolerance Test

To identify bile-tolerant strains, a bile tolerance test was conducted on the 40 strains that had previously demonstrated acid resistance and lactic acid production. All strains, except for one, grew on plates containing 1% oxgall (Difco^TM^ Oxgall, BD) (Table 1 and Figure 7c). The control strain, *L. acidophilus* KCTC 3111, also grew on MRS agar plates, confirming its bile tolerance.

#### 3.7.4. Heat Resistance Test

A heat resistance test was conducted on 39 strains that exhibited acid tolerance, lactic acid production, and bile tolerance (Table 1 and Figure 7d). Among these, 33 (84.61%), 32 (82.05%), and 28 strains (71.79%) survived at 40 °C, 50 °C, and 60 °C, respectively. The control strain, *L. acidophilus* KCTC 3111, also survived at all three temperatures, confirming its heat resistance.

#### 3.7.5. Dietary Enzyme of Protease

A protease activity test was conducted on 28 strains that exhibited acid tolerance, lactic acid production, bile tolerance, and heat resistance. The experiment aimed to identify strains capable of producing the digestive enzyme protease (Table 1 and Figure 7e). Among these 28 strains, 20 (71.43%) formed clear zones around their colonies, indicating protease production, while 8 strains (28.57%) did not. Among the 20 protease-producing strains, 12 (60%) produced large clear zones, indicating strong protease activity.

### 3.8. Identification of Candidate Probiotic Strains

In total, 109 candidate strains (JJ001−JJ109) were isolated from 40 canine faecal samples. Among these, 12 strains were selected for phenotypic characterisation (Table 2). All strains were Gram-positive, rod-shaped, and catalase-negative. To identify final probiotic candidates, an antibacterial activity test was conducted on the 12 selected strains (Figure 8). After co-culturing the candidate strains with three pathogenic bacteria—*E. coli* ATCC 35218, *S. aureus* ATCC 29213, and *S.* Typhimurium KCTC 2515—for 24 h, the results showed that five strains (JJ37, JJ68, JJ69, JJ71, and JJ77) exhibited bacterial counts below 10^8^ CFU/mL against *E. coli* and *S. aureus*. The control strain, *L. reuteri* KCTC3594, had a bacterial count of 10^8^ CFU/mL against the pathogenic strains, while the remaining strains exceeded 10^8^ CFU/mL. Against *S.* Typhimurium, all strains inhibited bacterial growth to below 10^1^ CFU/mL. Based on their antibacterial activity after 24 h of co-culture, the five strains (JJ37, JJ68, JJ69, JJ71, and JJ77) were identified as probiotic candidates.

### 3.9. Identification of L. reuteri JJ37, 68, 69, 71, and 77

The five selected strains were identified as *L. reuteri* JJ37, JJ68, JJ69, JJ71, and JJ77. Microgen analysis confirmed that these strains belonged to *L. reuteri* with over 99.9% similarity (Figure 9). The analysis results have been deposited in the GenBank database (*L. reuteri* JJ37, accession no. PV495746; *L. reuteri* JJ68, accession no. PV545326, *L. reuteri* JJ69, accession no. PV495749; *L. reuteri* JJ71, accession no. PV545322; and *L. reuteri* JJ77, accession no. PV545331). Moreover, the nucleotide sequencing chromatograms of the strains are shown in Appendix A.

## 4. Discussion

In this study, the gut microbiota of healthy puppies and adult dogs were analysed. Cultured microorganisms were then compared to identify and select *Lactobacillus* strains with probiotic potential. Faecal samples were collected from 40 companion dogs, with only one exhibiting signs of illness. Overall, clinical symptoms in both puppies and adult dogs were generally stable. The puppies weighed less than 2 kg (≤2 months old), while the adult dogs had a weight range of 2–7 kg (≥2 months old). Haematochezia was observed in one adult dog, but overall health remained stable. This study highlights the differences in gut microbiota composition between puppies and adult dogs and evaluates the potential for selecting *Lactobacillus* strains with antibacterial activity against pathogenic bacteria. This analysis is significant as it improves our understanding of gut microbiota changes related to age, weight, and health status. It can serve as foundational data for developing targeted probiotics for companion dogs.

In the phylum classification, no bacterial taxa were consistently abundant among puppies. However, in adult dogs, *Bacteroides* was the most prevalent taxon. The gut microbiota comprises over 1000 bacterial species, with *Bacteroidetes* and *Firmicutes* being the dominant phyla. In this analysis of three adult dogs and three puppies, the predominant bacterial taxa were *Proteobacteria*, *Firmicutes*, *Bacteroidetes,* and *Fusobacteria*. You and Kim et al. [11] examined the gut microbiota composition of 96 healthy dogs. Their phylum classification identifies *Proteobacteria*, *Firmicutes*, *Bacteroidetes*, and *Fusobacteria* as the predominant taxa, indicating that the results are statistically significant.

At the class level, no bacterial taxa were commonly abundant in puppies and adult dogs. Additionally, no bacterial taxa were consistently distributed among puppies. However, in adult dogs, *Bacteroidia* was the most abundant taxon, with dominant taxa including *Bacteroidales*, *Fusobacteria_c*, and *Clostridia*. In the phylum classification, no common taxa were observed in puppies, while adult dogs exhibited significant similarities. The bacterial taxa present in puppies included *Enterobacterales*, *Veillonellales*, and *Fusobacteriales*, but no single taxon was consistently abundant. In contrast, *Bacteroidales* was the most prevalent bacterial taxon in adult dogs. Puppies showed considerable variability in gut microbiota composition, whereas adult dogs maintained a more stable microbial composition across the phylum, class, and order levels. The gut microbiota composition of adult dogs aligns closely with findings from a previous study [11], reinforcing the consistency of these findings with other research.

This study was conducted to examine the changes in faecal microbiota composition over time across different growth stages in dogs. In this research, dogs younger than 2 months and those older than 12 months were categorised into two groups to observe variations in faecal microbiota. Previous studies confirm the faecal microbiota composition in both puppies and adult dogs. At the phylum level, bacterial taxa exhibited a significant difference of more than 1.5-fold (>1.5%), including *Bacteroidetes*, *Fusobacteria,* and *Proteobacteria*. At the class level, the identified taxa were *Bacilli*, *Bacteroidia*, *Clostridia*, *Epsilonproteobacteria*, *Fusobacteria_c*, *Gammaproteobacteria,* and *Negativicutes*. The order-level classification included *Acidaminococcales*, *Bacteroidales*, *Clostridiales*, *Enterobacterales*, *Fusobacteriales*, *Lactobacillales*, *Pseudomonadales*, and *Veillonellales*. Significant differences in microbial diversity and abundance were observed between pre-weaning and post-weaning stages, as well as among puppies, adult dogs, and elderly dogs [12,13]. Previous studies show that faecal microbial composition changes with developmental stages in both humans and pigs. In humans, the faecal microbiota of infants becomes similar to that of adults once they transition to an adult-like diet, indicating that microbial communities are influenced by diet and environmental factors [14]. In pigs, the microbial population increases until approximately 20 weeks after birth, after which no major changes are observed [12].

In puppies, the microbial composition is predominantly characterised by *Lactobacillales*, while adult dogs exhibit a predominance of *Bacteroidetes*. The genus *Lactobacillus*, made up of lactic acid bacteria, is more prevalent in puppies. It plays a crucial role in lactose digestion during the milk consumption phase [15]. However, following weaning and dietary transition over the first year, its relative abundance decreases, as indicated by taxonomic classification. In contrast, adult dogs show a higher abundance of *Bacteroidetes*, a dominant bacterial group in the gastrointestinal tracts of animals and are primarily involved in polysaccharide metabolism [16]. These findings suggest that, as dogs mature, their carbohydrate intake increases, leading to a proportional rise in *Bacteroidetes* abundance.

The gut microbiota maintains a stable community by balancing beneficial and harmful bacteria. Beneficial bacteria stimulate the immune function of the host and prevent various bacterial infections, while harmful bacteria induce putrefaction and produce carcinogens or toxins. For optimal gut health, beneficial bacteria must predominate [17]. Within the biological classification system, *Firmicutes* and *Bacteroidetes* are the primary constituents of the bacterial species present. A previous study shows that probiotic supplementation improves health status, resulting in a decrease in *Firmicutes* from 46–31% and an *increase* in *Bacteroidetes* from 22–28% [18]. These findings suggest that improved health is associated with higher *Bacteroidetes* and lower *Firmicutes* levels.

An analysis of the gut microbiota composition in puppies and adult dogs reveals that puppies have a lower relative abundance of *Bacteroidetes* and a higher relative abundance of *Firmicutes* than adult dogs. This suggests that adult dogs have a more balanced gut microbiome. The proportional changes in *Firmicutes* and *Bacteroidetes*, along with the F/B ratio, serve as indicators of gut microbial health [19]. The lower F/B ratio observed in adult dogs indicates that they maintain a more stable gut microbiota.

Blood agar was utilised to isolate a broad spectrum of pathogens that are challenging to culture and to evaluate the haemolytic activity of isolates [20]. Haemolytic activity was detected in only two adult dogs, with no haemolytic isolates identified in other samples. TSN agar was employed to culture and isolate *Clostridium perfringens* [21]. This bacterium secretes toxins and enzymes responsible for various gastrointestinal diseases, including food poisoning, non-foodborne diarrhoea, and enteritis in humans and animals [22]. PDA was used for the cultivation of fungi and moulds [23]. MC facilitated the isolation and differentiation of pathogenic enteric bacteria. SS agar was employed to selectively isolate and differentiate *Salmonella* and *Shigella*. MS agar enables the selective cultivation and isolation of *S. aureus*. BG agar was used for the selective isolation of *Salmonella* spp., excluding *S.* Typhimurium. Additionally, MACS was specifically used to culture enterohaemorrhagic *E. coli*. MRS agar supported the isolation of *Lactobacillus* spp., while TOS selectively promoted the growth of *Bifidobacterium* spp. by inhibiting *Lactobacillus* growth. PCA was employed to quantify LAB. PCA, MRS, and TOS, which are associated with LAB, exhibited higher microbial counts in puppies. However, no statistically significant differences were observed between puppies and adult dogs in any medium except MS agar. This finding aligns with that of previous research, which reports no significant differences in faecal microbial community distribution patterns between puppies and adult dogs across various media types [24].

LAB must survive passage through the stomach, where the pH is ≤3, and reach the small intestine to exert their physiological functions [25]. A previous study reports a 40.6% survival was observed when 101 candidate LAB strains isolated from dogs were cultured at pH 2.5 for 3 h [26]. In the present study, survival was 51.4% when cultured at pH 2 for 1.5 h. Although this difference was not statistically significant, the survival rate consistently remained approximately 50%.

LAB naturally produce lactic acid [27], which exerts bactericidal effects against susceptible bacteria, thus inhibiting harmful gut microorganisms [28]. Therefore, acid-producing strains are expected to have a positive effect.

To reach the intestine, ingested LAB must pass through the stomach, pancreas, and duodenum. Bile tolerance in these areas is a critical characteristic [29]. In this study, when cultured for 2 days in media with 1% bile, all strains, except for one, survived. Similarly, a previous study reports a 99% survival rate when LAB were cultured for 2 h in media with 1% bile, with only one strain failing to survive [26], which aligns with our findings.

Probiotics are typically administered directly or mixed with feed in the form of powder, pellets, granules, pills, or paste. This processing usually involves heat treatment at temperatures ranging from 50–80 °C [30]. A previous study shows that probiotic strains isolated from dogs exhibit a 92.5% survival rate after being cultured at 50 °C for 5 min, while survival decreases to 80% when cultured at 60 °C for 5 min [26]. In the present study, the survival rate was 71.8% when cultured at 50 °C for 1 h, and the same survival rate was observed at 60 °C for 1 h. The lower survival rate in this study is attributed to the extended culturing time compared to that of the previous study.

Probiotics have a positive effect on protein digestion, influencing digestion and nutrient absorption in the gastrointestinal tract. They induce the activity of digestive proteolytic enzymes and peptidases in the host, and some strains also release enzymes involved in protein breakdown [31]. A previous study characterised LAB strains selected for probiotic use, identifying them as Gram-positive, catalase-negative, non-motile, non-spore-forming, and rod-shaped *bacilli* [32]. The findings of this present study are similar to those of the previous research, supporting the assessment of candidate probiotic strains. LAB produce lactic acid as a metabolic by-product, lowering pH [33], which effectively inhibits the growth of pathogenic bacteria through various antimicrobial mechanisms [34]. Although studies show that increasing LAB populations reduce the growth of pathogenic bacteria such as *S. aureus*, *E. coli*, and *Salmonella*, research on their specific effects remains limited [35].

With the increase in pet ownership in Korea, pets have become integral family members, raising concerns about health issues, particularly the transmission of infectious diseases between pets and their owners. As of 2023, the number of pet-owning households in Korea has increased approximately 2.8 times compared to that of 2020, with dogs comprising 75.6% of the total pet population [36]. However, the shared living environment between pets and humans increases the risk of the transmission of infectious diseases such as pathogenic bacteria. A representative zoonotic disease is ringworm, a fungal infection [37] that spreads from pets to humans through direct contact with fur or skin, posing a higher risk to vulnerable groups such as the elderly and children. Additionally, *Salmonella* can be transmitted through pet faeces, causing severe gastroenteritis and, in some cases, septicemia [38]. *Salmonella* infection is a significant public health issue, posing a direct risk to both pet and human health. Moreover, *E. coli* infection is another concern, as pathogenic *E. coli* from pets can be transmitted to humans, causing symptoms such as gastroenteritis, abdominal pain, and diarrhoea [39]. The emergence of antibiotic-resistant *E. coli* strains exacerbates treatment difficulties and is recognised as a critical public health concern. Other bacterial infections, such as *campylobacteriosis*, can also spread between pets and humans, typically causing diarrhoea, fever, and abdominal pain, resulting in severe complications in immunocompromised individuals [40]. To address the issue of infectious disease transmissions between pets and their owners, probiotics have gained attention as a potential alternative to antibiotics. *L. reuteri* is a well-studied lactic acid bacterium with demonstrated antimicrobial activity, especially in inhibiting *Salmonella* and *E. coli* [41].

## 5. Conclusions

In this study, through 16S rRNA gene sequencing and phylogenetic analysis, we confirmed that strains such as JJ37, JJ68, JJ69, JJ71, and JJ77 exhibit significant potential in inhibiting pathogenic bacteria. These findings provide an essential foundation for zoonotic disease prevention and health management. Moreover, the growth of the pet market has led to the commercialisation of probiotics in specialised pet foods and canine health supplements. This contributes to improved quality of life for pets and their owners. Future research should focus on developing customised probiotics tailored to breed, health status, and disease specificity in dogs. Such advancements may contribute to the prevention of pathogenic bacteria infections and improve the balance of gut microbiota in pets.

## Figures and Tables

**Figure 1 microorganisms-13-01059-f001:**
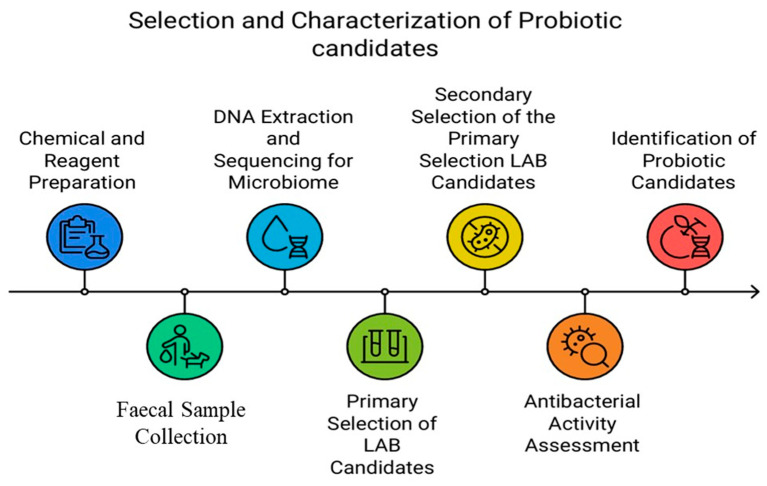
Process of the experiment for probiotic candidate selection. Abbreviation: LAB, lactic acid bacteria.

**Figure 2 microorganisms-13-01059-f002:**
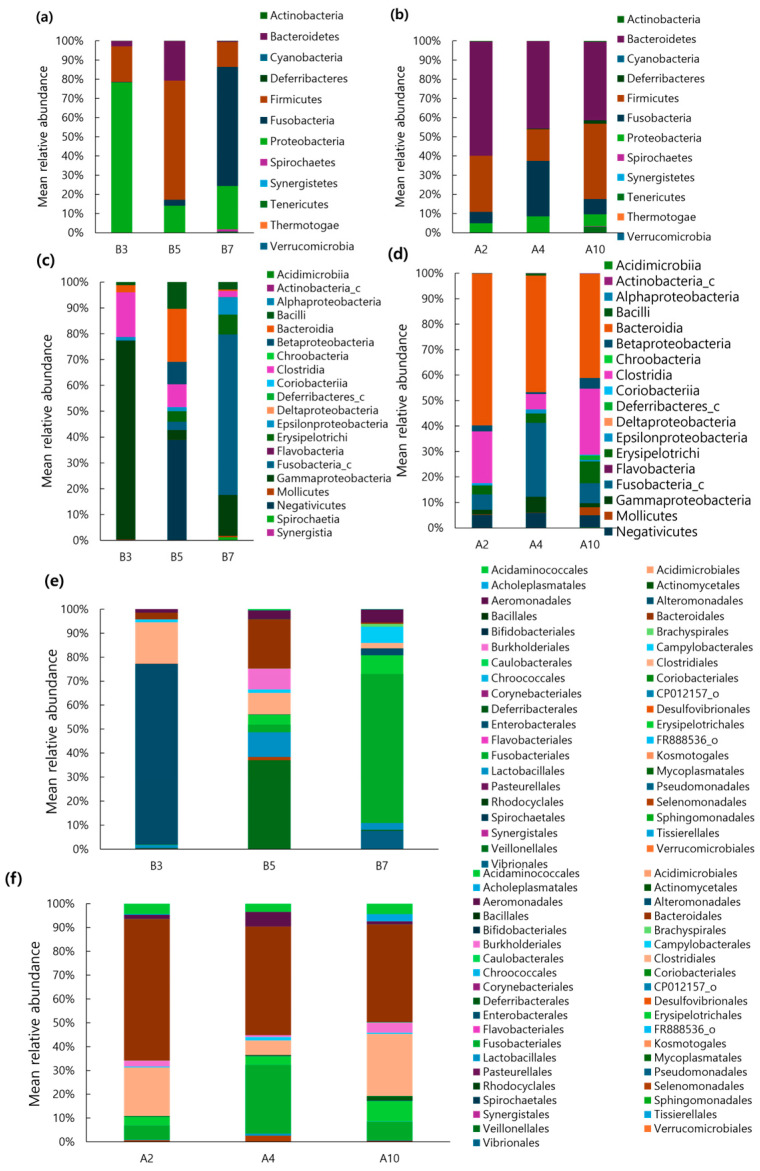
Taxonomic classification of 40 faecal samples from healthy dogs, including adult dogs and puppies. (**a**,**b**) Phylum level, (**c**,**d**) class level, and (**e**,**f**) order level.

**Figure 3 microorganisms-13-01059-f003:**
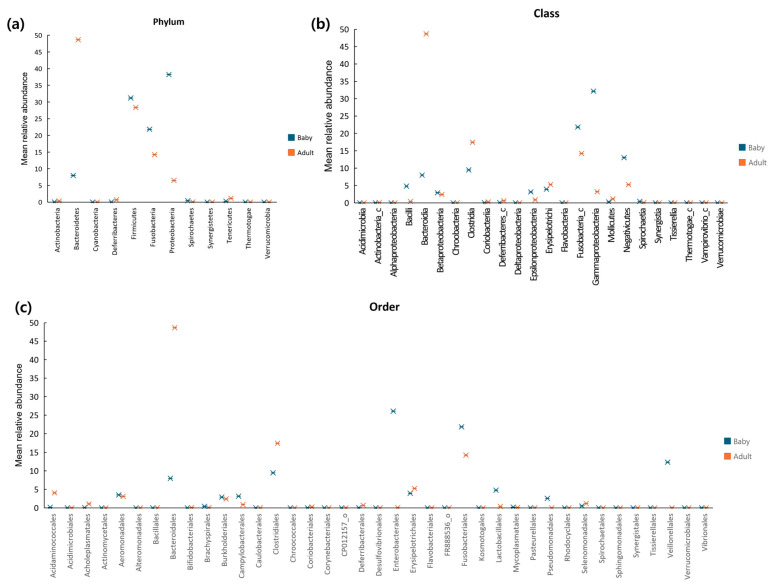
Correlation between bacterial taxa and age group (adult dogs vs. puppies) at the phylum level (**a**), class level (**b**), and order level (**c**).

**Figure 4 microorganisms-13-01059-f004:**
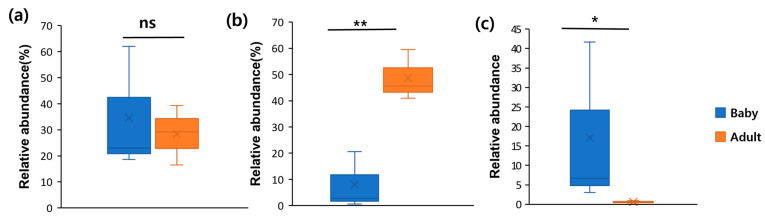
The relative abundances of *Firmicutes* (**a**), *Bacteroidetes* (**b**), and variability in the F/B ratio (**c**) in the gut microbiota of adult dogs and puppies. Box plots were constructed using R. * *p* < 0.05 and ** *p* < 0.01. Abbreviations: F/B, *Firmicutes*/*Bacteroidetes*; ns, non-significant.

**Figure 5 microorganisms-13-01059-f005:**
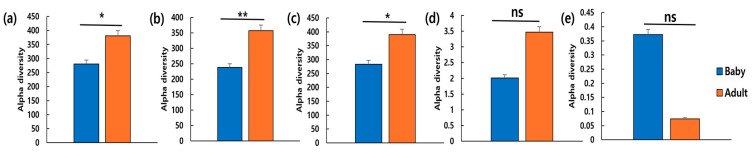
Differences in alpha diversity between puppies (*n* = 3) and adult dogs (*n* = 3). Species richness reflected via (**a**) ACE, (**b**) Chao1, (**c**) Jackknife, (**d**) Shannon, and (**e**) Simpson indices. * *p* < 0.05 and ** *p* < 0.01. Abbreviation: ns, non-significant.

**Figure 6 microorganisms-13-01059-f006:**
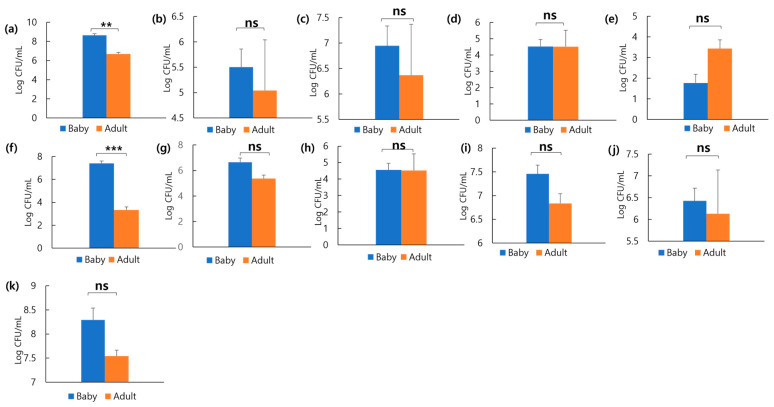
Diversity and strain-specific characteristics of gut microbiota, as revealed by culturing faecal samples on various media. Bacterial colony counts on (**a**) Blood agar, (**b**) TSN, (**c**) PDA, (**d**) MC, (**e**) SS, (**f**) MS, (**g**) BG, (**h**) MACS, (**i**) MRS, (**j**) TOS, and (**k**) PCA with BCP media of adult dogs and puppies. ** *p* < 0.01 and *** *p* < 0.001. Abbreviations: TSN, tryptone sulfite neomycin; PDA, potato dextrose agar; MC, MacConkey; SS, Salmonella–Shigella; MS, mannitol salt; BG, Brilliant Green; MACS, MacConkey sorbitol; MRS, De Man, Rogosa, and Sharpe; TOS, transgalactosylated oligosaccharides with MUP; PCA, plate count agar; BCP, Bromocresol Purple; ns, non-significant.

**Figure 7 microorganisms-13-01059-f007:**
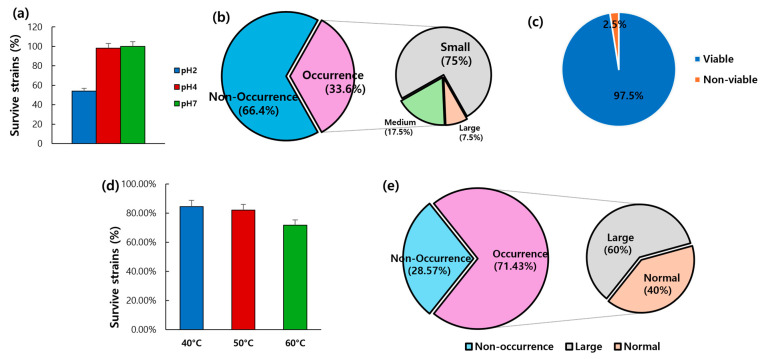
Probiotic characteristics of selected LAB from the faeces of dogs. (**a**) Acid tolerance, (**b**) Lactic acid production, (**c**) Bile resistance, (**d**) Heat resistance, and (**e**) Dietary enzyme of protease. Abbreviation: LAB, lactic acid bacteria.

**Figure 8 microorganisms-13-01059-f008:**
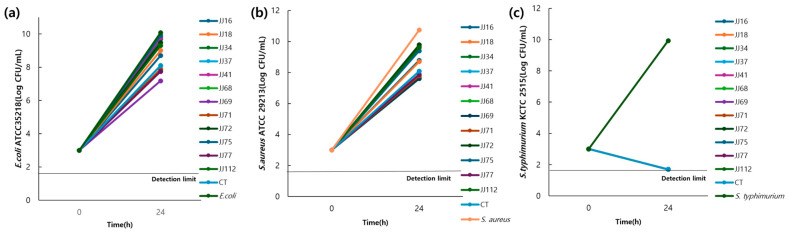
Co-culture of *Lactobacillus* with enterotoxigenic pathogens (**a**) *E. coli* ATCC 35218, (**b**) *S. aureus* ATCC 29213, and (**c**) *Salmonella* Typhimurium KCTC 2515. The CFU of enterotoxigenic pathogens at 10^3^ CFU/mL after co-culture with *Lactobacillus* (10^9^ CFU/mL) for 24 h. Abbreviations: *E. coli*, *Escherichia coli*; *S. aureus*, *Staphylococcus aureus*; CFU, colony-forming units; CT, control.

**Figure 9 microorganisms-13-01059-f009:**
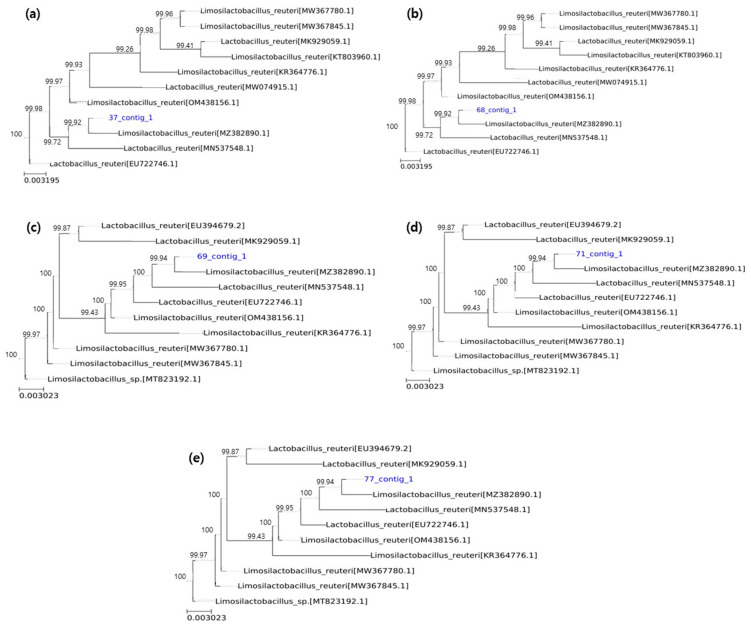
Phylogenetic tree of *L. reuteri* strains (**a**) *L. reuteri* JJ37 (37_contig_1), (**b**) *L. reuteri* JJ68 (68_contig_1), (**c**) *L. reuteri* JJ69 (69_contig_1), (**d**) *L. reuteri* JJ71 (71_contig_1), and (**e**) *L. reuteri* JJ77 (77_contig_1).

**Table 1 microorganisms-13-01059-t001:** Probiotic characteristics of selected lactic acid bacteria from the faeces of dogs.

Characteristics		Number of Isolates (%)
Acid tolerance	pH 2, 1.5 h	59/109 (54.12)
	pH 4, 1.5 h	107/109 (98.17)
	pH 7, 1.5 h	109/109 (100.0)
Lactic acid production	+	30/119 (25.21)
	++	7/119 (5.88)
	+++	3/119 (2.52)
Bile tolerance	1.0% oxgall, 48 h	39/40 (97.5)
Heat resistance	40 °C, 1 h	33/39 (84.61)
	50 °C, 1 h	32/39 (82.05)
	60 °C, 1 h	28/39 (71.79)
Dietary enzyme of protease	+	8/28 (28.57)
	++	12/28 (60.0)

+, Minimum; ++, Medium; +++, Maximum. Classified on lactic acid production (brightness: weak, moderate, and strong translucent ring) and protease (diameter: 0.5–1 cm, 1.0–1.5 cm, and ≥1.5 cm).

**Table 2 microorganisms-13-01059-t002:** Characterisation of lactobacilli isolated from dog faeces.

Isolates	Gram Staining	Cell Morphology	Catalase	Antibacterial Activity
JJ16	P	Rod	N	+
JJ18	P	Rod	N	+
JJ34	P	Rod	N	+
JJ37	P	Rod	N	+++
JJ41	P	Rod	N	+
JJ68	P	Rod	N	+++
JJ69	P	Rod	N	+++
JJ71	P	Rod	N	+++
JJ72	P	Rod	N	+
JJ75	P	Rod	N	+
JJ77	P	Rod	N	+++
JJ112	P	Rod	N	+
CT	P	Rod	N	+

P, Positive; N, Negative; +++, Maximum; +, Minimum. CT, Control.

## Data Availability

All data generated for this study are contained within the article/Appendix A.

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
