# Peer review of "Microbiota Analysis and Characterisation of the Novel Limosilactobacillus Strains Isolated from Dogs"

_microorganisms, 2025, doi:10.3390/microorganisms13051059_

Round 1

Reviewer 1 Report

Comments and Suggestions for Authors

In this study, authors isolated lactic acid bacteria (LAB) strains from dog fecal samples (puppies: < 2 months old and adult dogs: > 12 months old)) and assessed their potential to inhibit E. coli, S. aureus, and Salmonella spp. In total, 109 primary LAB candidates were isolated from fecal samples and 5 LAB candidates with probiotic potential such as acid tolerance, bile salt resistance, acid production, heat resistance, protease activity, and antimicrobial activity against E. coli, S. aureus, and Salmonella spp. were screened. All five strains were Lactobacillus reuteri, with rJJ37 and JJ69 are considered as the final probiotic candidates.

My comments are below:

  • A supplementary file reporting the first screening of the 109 isolates is recommended.
  • Please provide a clearer figure 6.
  • Using only 16S for identification is not enough, you should add minimum two more housekeeping genes such as recA and atpD or others, and concatenate them with the 16S.
  • Please provide the 16S PCR gel and the chromatograms received from the sequencer.
  • Why only two strains were concerned by the phylogeny, you should analyze the five LAB isolated strains in one single tree, no need to analyze each strain separately?
  • For phylogeny, you should use an outgroup strain and add the bootstrap values to get a clear idea.
  • The accession number of all the 16S sequences should be provided, no need to add the sequences in the supplementary file. Accession numbers are enough.

Author Response

Response to Reviewer 1 Comments

1. Summary

2. Questions for General Evaluation

Reviewer’s Evaluation

Response and Revisions

Does the introduction provide sufficient background and include all relevant references?

Yes/Can be improved/Must be improved/Not applicable

We have revised the whole manuscript based on the reviewers' comments, including double-checking the relevancy of all references.

Are all the cited references relevant to the research?

Yes/Can be improved/Must be improved/Not applicable

Is the research design appropriate?

Yes/Can be improved/Must be improved/Not applicable

Are the methods adequately described?

Yes/Can be improved/Must be improved/Not applicable

Are the results clearly presented?

Yes/Can be improved/Must be improved/Not applicable

Are the conclusions supported by the results?

Yes/Can be improved/Must be improved/Not applicable

3. Point-by-point response to Comments and Suggestions for Authors

Comment 1: A supplementary file reporting the first screening of the 109 isolates is recommended.

Response 1: Thank you. We selected an initial 109 strains based on Gram staining and morphology. Among them, 12 isolates were selected on the basis of catalase-negative and antibacterial activity. We have mentioned this in the Results section (subsection-3.8. Identification of candidate probiotic strains).

Comment 2: Please provide a clearer figure 6.

Response 2: Thank you. We have provided the new Figure 6 and separate plate images, as shown in Supplementary Figure S2.  

Comment 3: Using only 16S for identification is not enough, you should add minimum two more housekeeping genes such as recA and atpD or others, and concatenate them with the 16S.

Response 3: Thanks for your comments. In this study, we have performed 16S rRNA analysis to identify the bacterial strain. In our previous study (Ali et al., 2022), we conducted a similar experiment to identify Limosilactobacillus. It would be better to do more experiments related to the identification, but the limited time for review may not be enough to do more experiments. We will consider performing this experiment in our forthcoming project.

Comments 4: Please provide the 16S PCR gel and the chromatograms received from the sequencer.

Response 4: Thanks for your comments. We have provided the chromatograms of the five strains (JJ37, JJ68, JJ69, JJ71, and JJ77) provided by Macrogen (Macrogen Inc., Seoul, Korea).

We have added the following sentence in the revised manuscript (section-Results; subsection-3.9. Identification of L. reuteri JJ 37, 68, 69, 71, and 77).

“Moreover, the nucleotide sequencing chromatograms of the strains are shown in Supplementary Figures S3-S7.

Comments 5: Why only two strains were concerned by the phylogeny, you should analyze the five LAB isolated strains in one single tree, no need to analyze each strain separately?

Response 2: Thanks for your comments. We have provided phylogeny of five strains (JJ37, JJ68, JJ69, JJ71, and JJ77).

Comments 6: For phylogeny, you should use an outgroup strain and add the bootstrap values to get a clear idea.

Response 6: Thanks for your comment. We have performed the experiments to identify the strains by Macrogen (Macrogen Inc., Seoul, Korea). The strains' similarity was determined by their nucleotide sequence, based on scales of 0.003195 for JJ37 and JJ68, and 0.003023 for JJ69, JJ71, and JJ77 (Figure 9).

Comment 7. The accession number of all the 16S sequences should be provided, no need to add the sequences in the supplementary file. Accession numbers are enough.

Response 7: Thanks for your comments. We provided the accession number of the nucleotide sequence of the five strains in the revised manuscript section-Results; subsection-3.9. Identification of L. reuteri JJ 37, 68, 69, 71, and 77). We removed this supplementary figure).

“The analysis results have been deposited in the GenBank database (L. reuteri JJ37, accession no. PV495746; L. reuteri JJ68, accession no. PV545326; L. reuteri JJ69, accession no. PV495749; L. reuteri JJ71, accession no.PV545322; and L. reuteri JJ77, accession no.PV545331).”

Response to Comments on the Quality of English Language

Point 2: The English is fine and does not require any improvement.

References

Ali MS, Lee E-B, Quah Y, et al. Heat-Killed Limosilactobacillus Reuteri PSC102 Ameliorates Impaired Immunity in  Cyclophosphamide-Induced Immunosuppressed Mice. Front Microbiol 2022;13:820838; doi: 10.3389/fmicb.2022.820838.

Reviewer 2 Report

Comments and Suggestions for Authors

This article provides information on the microbiota analysis and characterisation of the novel Limosilactobacillus strains isolated from dogs. It is in general appropriately organized, carried out and written, however there are some points that should be corrected or clarified. Please check comments and corrections in the attached file.

Author Response

Response to Reviewer 2 Comments

1. Summary

2. Questions for General Evaluation

Reviewer’s Evaluation

Response and Revisions

Does the introduction provide sufficient background and include all relevant references?

Yes/Can be improved/Must be improved/Not applicable

We have revised the whole manuscript based on the reviewers' comments, including double-checking the relevancy of all references.

Are all the cited references relevant to the research?

Yes/Can be improved/Must be improved/Not applicable

Is the research design appropriate?

Yes/Can be improved/Must be improved/Not applicable

Are the methods adequately described?

Yes/Can be improved/Must be improved/Not applicable

Are the results clearly presented?

Yes/Can be improved/Must be improved/Not applicable

Are the conclusions supported by the results?

Yes/Can be improved/Must be improved/Not applicable

3. Point-by-point response to reviewer 1.

Comment 1: Line 28. What do you mean “it”?

Response 1: Thanks for your comments. We revised as follows in the revised manuscript (Last sentence of the abstract).

“They promote healthier cohabitation between dogs and their owners.”

Comment 2: Line 61-64. Please merge with the previous sentence. Both sentences refer to the same.

Response 2: Thanks for your comments. We have modified this by adding the following information to the revised manuscript (Introdecution-3rd paragraph).

“Moreover, as conventional antibiotic therapy remains limited, the growing demand for alternative treatments, including targeted probiotic formulations, emphasizes the need for continued research into their specific mechanisms and benefits, representing a critical advancement in the animal medicine sector.”

Comment 3: Line 71-73. This part is not necessary here.

Response 3: Thanks for your comments and suggestions. We have removed this part from the revised manuscript. 

Comments 4: Line 242. “was observed”

Response 4: Thank you. We corrected this in the revised manuscript (section-Results; subsection-3.2. Metagenomic analysis of the gut microbiota in puppies and adult dogs-2nd paragraph).

Comment 5. Line 242. Do you mean "Bacteroidia"?

Response 5: Thanks for your comments. We have corrected this as follows in the revised manuscript (section-Results; subsection-3.2. Metagenomic analysis of the gut microbiota in puppies and adult dogs-2nd paragraph).

“In adult dogs, A2 was dominated by Bacterodia (59.48%) and Clostridia (20.34%), A4 by Bacteroidia (45.61%) and Fusobacteria (28.88%), and A10 by Bacteroidia (40.9%) and Clostridia (25.97%).”

Comment 6. Line 245-246. Repetition, if L242-243 remain

Response 6. Thank you very much for your comments and suggestions. We have corrected this as follows in the revised manuscript (section-Results; subsection-3.2. Metagenomic analysis of the gut microbiota in puppies and adult dogs-2nd paragraph).

“In adult dogs, A2 was dominated by Bacterodia (59.48%) and Clostridia (20.34%), A4 by Bacteroidia (45.61%) and Fusobacteria (28.88%), and A10 by Bacteroidia (40.9%) and Clostridia (25.97%).”

Comment 8. Line 254. What about adult dogs?

Response 8. Thanks for your comments. We have added the following sentence in the revised manuscript (section-Results; subsection-3.2. Metagenomic analysis of the gut microbiota in puppies and adult dogs-3rd paragraph).

“In adult dogs, A2 exhibited the highest proportion of Bacteroidales (59.47%), followed by Clostridiales (20.34%) and Fusobacteriales (5.96%); A4 predominantly contained Bacteroidales (45.61%) followed by Fusobacteriales (28.86%) and Aeromonadales (6.2%); and A10 was dominated by Bacterodiales (40.9%) followed by Clostridiales (25.96%) and Erysipelotichales (8.57%). A consistent dominance of Bacteroidales (>40%) was observed across all adult dogs.”

Comment 9. Line 272-274. Other classes are shown in Fig. 3B: Bacilli, Bacteroidia, Betaproteobacteria, Clostridia, Epsilonproteobacteria, Erysipelotrichi, Fusobacteria_c, Gammaproteobacteria, Negativicutes.

Response 9. Thanks for your comments. We have rearranged this in the revised manuscript (section-Results; subsection-3.3. Differences in Gut microbial composition between puppies and adult dogs-2nd paragraph).

Bacilli, Bacteroidia, Betaproteobacteria, Clostridia, Epsilonproteobacteria, Erysipelotrichi, Fusobacteria_c, Gammaproteobacteria, and Negativicutes.”

Comment 10. Line 275-277. Please also check

Response 10. We have rearranged this in the revised manuscript (section-Results; subsection-3.3. Differences in Gut microbial composition between puppies and adult dogs-2nd paragraph).

“In contrast, adult dogs exhibited higher proportion of Bacteroidia, Betaproteobacteria, Clostridia, Erysipelotrichi, Fusobacteria_c, Gammaproteobacteria, and Negativicutes.”

Comment 11. Line 278. Please also check

Response 11. Thank you. Eplionproteobacteia” will be Epsilonproteobacteria,” We have corrected this in the revised manuscript (section-Results; subsection-3.3. Differences in Gut microbial composition between puppies and adult dogs-2nd paragraph).

Comment 12. Line 289. What about "Bacteroidales", "Clostridiales", "Erysipelotrichales"?

Response 12. Thanks for your comment. We have corrected this in the revised manuscript (section-Results; subsection-3.3. Differences in Gut microbial composition between puppies and adult dogs-3rd paragraph).

Comment 13. Line 318-319. In Fig. 4C, a significantly lower percentage is shown for adults (<1)!

Response 13. Thanks for your comment. We have corrected this in the revised manuscript (section-Results; subsection- 3.4. F/B ratio in adult dogs and puppies and Figure 4).

Comment 14. Line 321-323. Different values are shown in Fig. 4A

Response 13. Thanks for your comment. We have corrected this in the revised manuscript (section-Results; subsection- 3.4. F/B ratio in adult dogs and puppies and Figure 4).

Comment 15. Line 348-349. Other values are presented in Fig. 6A

Response 15. Thanks for your comments. We corrected this in the revised manuscript (section-Results; subsection- 3.6. Comparison of faecal microbial distribution between puppies and adult dogs). Moreover, we have separated the plate image as Supplementary Figure S2 for better vision).

Comment 16. Line 350-351. Please check Fig. 6B

Response 16. We corrected this in the revised manuscript (section-Results; subsection- 3.6. Comparison of faecal microbial distribution between puppies and adult dogs). Moreover, we have separated the plate image as Supplementary Figure S2 for better vision).

Comment 17. Line 351-357. Please check values in Fig. 6

Response 17. We corrected this in the revised manuscript (section-Results; subsection- 3.6. Comparison of faecal microbial distribution between puppies and adult dogs). Moreover, we have separated the plate image as Supplementary Figure S2 for better vision).

Comment 18. Line 358-365. Please check values in Fig. 6

Response 18. We corrected this in the revised manuscript (section-Results; subsection- 3.6. Comparison of faecal microbial distribution between puppies and adult dogs). Moreover, we have separated the plate image as Supplementary Figure S2 for better vision).

Comment 19. Line 383. or 51.4% (Table 1)?

Response 19. Thank you. We have corrected this in the revised manuscript (section-Results; subsection-3.7. (3.7.1.) and Table 1).

Comment 20. Line 410. 42.85% is shown in Fig.7e

Response 20. Thank you. We have corrected this in the revised manuscript (Fig.7e).

Comment 21. Line 411. or 42.85%? (Fig. 7e)

Response 21. Thank you. We have corrected this in the revised manuscript (Fig.7e).

Comment 22. Line 521-522. This is not shown in Figure 4

Response 22. We have corrected this in the revised manuscript (section-Results; subsection- 3.4. F/B ratio in adult dogs and puppies and Figure 4).

Response to Comments on the Quality of English Language

Point 2: The English is fine and does not require any improvement.

Round 2

Reviewer 2 Report

Comments and Suggestions for Authors

Authors made the necessary amendments and I suggest the acceptance of their article

Author Response

Comments: Authors made the necessary amendments and I suggest the acceptance of their article.

Response: Thank you very much for your time and efforts.